# Rust Prevention Property of a New Organic Inhibitor under Different Conditions

**DOI:** 10.3390/ma17092168

**Published:** 2024-05-06

**Authors:** Xingxing Guo, Chengsheng Wang, Hua Fu, Li Tian, Hua Song

**Affiliations:** 1Department of Civil Engineering, Changzhi Vocational and Technical College, Changzhi 046000, China; 13485385732@163.com; 2Tianjin Port Engineering Design & Consulting Company Ltd. of CCCC, Tianjin 300456, China; wcs90@126.com; 3School of Civil Engineering, Qingdao University of Technology, Qingdao 266033, China; fs215379@163.com (H.F.); carrie_song@icloud.com (H.S.)

**Keywords:** inhibitor, weight loss, electrochemical impedance spectroscopy, potentiodynamic polarization

## Abstract

The corrosion resistance properties of a new type of environmentally-friendly organic inhibitor containing amino ketone molecules are presented in this paper. To evaluate the prevention effect of the inhibitor on corrosion of reinforcement, the electrochemical characteristics of steels in the simulated concrete pore solution (SPS) were investigated under varied conditions of the relevant parameters, including concentrations of the inhibitor and NaCl, pH value, and temperature. The inhibition efficiency of the material was characterized through electrochemical impedance spectroscopy (EIS), potentiodynamic polarization, and the weight loss of steels. The results reveal a significant improvement in the corrosion resistance of steels with the inhibitor. A maximum resistance value of 89.07% was achieved at an inhibitor concentration of 4%. Moreover, the new organic inhibitor exhibited good corrosion protection capability for steels under different NaCl concentrations. Its inhibition efficiency was determined to be 65.62, 80.06, and 66.30% at NaCl concentrations of 2, 3.5 and 5%, respectively. On the other hand, it was found that an alkaline environment was favorable for an enhanced corrosion prevention effect, and an optimal pH value of 11.3 was observed in this work. Besides, the inhibition efficiencies at different temperatures showed a trend of 25 > 35 > 40 > 20 > 30 °C, with a maximum value of 81.32% at 25 °C. The above results suggest that the new organic material has high potential to be used as an eco-friendly and long-term durable inhibitor for steel corrosion prevention under complex conditions.

## 1. Introduction

Reinforced concrete has been widely used in various construction projects due to its economical features, practicality, and durability [1,2]. The durability of reinforced concrete structures has been receiving increased worldwide attention because durability problems can cause heavy casualties and serious economic loss. There are many intrinsic factors affecting concrete corrosion, such as the concrete structure type and property, the construction quality, and the cover thickness [3]. In addition, problems and failures of reinforced concrete structures could also be caused by other various factors, including steel corrosion, carbonation, freeze-thaw damage, chemical erosion, and alkali aggregate reactions, among which steel corrosion is a major reason [4,5,6]. Steels are susceptible to corrosion in aggressive media containing chloride, sulfate, or nitrate ions, especially in the marine environment [7]. Through diffusion and infiltration, chloride ions in seawater can transfer into concrete and accumulate on the surface of steel bars [8,9]. When the ion concentration reaches the specific threshold value on the steel bar surface, coupled with sufficient oxygen and water supply, the passive film on steel bars generated in a strong alkaline environment could be destroyed, which leads to so-called pitting corrosion [2,10]. With the continuation of steel corrosion, the concrete cover around steel bars may crack and detach from the structure, resulting in a decrease in the bearing capacity of the material [11]. Therefore, it is crucial to urgently and effectively prevent steel corrosion in the field of civil engineering [12,13].

A variety of approaches have been widely applied to protect steel bars from corrosion, for example, technologies of cathodic protection [14], conversion film [15,16] and coating [17,18], as well as the use of corrosion inhibitors. Cathodic protection is an effective way to decrease steel corrosion [19,20]. However, it is difficult to carry out large-scale cathodic protection for steel bars, due to the complexity of the technology construction and operation in concrete structures [21,22,23]. Similarly, the conversion film technique also has limited application for the protection of steel bars in concrete. Coating on steel bar surfaces can inhibit corrosion, but the presence of the coating layers greatly reduces the bonding behavior between steel bars and concrete, leading to a decrease in the overall bearing capacity of the material. In addition, during construction works, the coatings could be easily damaged and peeled off, forming a status of small anodes and large cathodes, therefore accelerating the steel corrosion process [24]. Generally speaking, high-performance corrosion inhibitors, combined with their low cost, easy operation, and high safety characteristics, have been determined to be a promising solution for preventing corrosion of steel bars in concrete.

Various inorganic inhibitors of alkalis, chromates, nitrite, phosphates, hypophosphates, and fluorides, as well as organic inhibitors, have been widely used in modern civil engineering areas. Several studies have reported that the application of inorganic inhibitors is limited by their serious environmental pollution, whereas the corrosion resistance efficiency of organic inhibitors varies greatly depending on the working environment [25,26]. Ma et al. [27] introduced a thiadiazole-derivative inhibitor with high inhibition efficiencies of 78.73–95.67%, and the maximum value was achieved with an inhibitor concentration of 100 mg/L. Besides, in the investigated pH range of 5.5–9.5, a gradual increase in the inhibition efficiency with the pH value was observed. Li et al. [28] showed that the corrosion resistance of a mixed-type inhibitor (myclobutanil and hexaconazole) for copper material decreases with a pH value order of 7.5 > 6.5 > 8.5 > 5.5, which reveals a better corrosion inhibition capability of the compounds in a near neutral condition, than in acidic or alkaline conditions. With respect to the influence of the temperature, it was found that although the temperature increase restricts the adsorption of the myclobutanil and hexaconazole inhibitor compounds, the charge transfer process in corrosion could still be decreased greatly at elevated temperatures. Tian reported five environmentally-friendly inhibitors with excellent corrosion inhibition effects, which showed a negative linear function between corrosion resistance efficiency and temperature.

In this work, the corrosion inhibition properties and mechanism of a new environmentally-friendly organic inhibitor containing amino ketone molecules were studied. The effect of the inhibitor on concrete corrosion was evaluated by analyzing the electrochemical characteristics of steels in the simulated concrete pore solution under different inhibitors and NaCl concentrations, pH values, and temperatures.

## 2. Synthesis of the Inhibitor

The organic rust inhibitor used in the test contains the amino ketone molecules made by the chemical reaction of ethanol, dimethylamine, formaldehyde, acetophenone, and other substances, and its molecular generation formula is shown in Figure 1.

The amino ketone inhibitor has the following characteristics:The nitrogen and oxygen atoms of the carbonyl group in the amino ketone molecule are all electronegative atoms, which can effectively adsorb on the oxide film at the reinforced steel surface;The amide and carbonyl groups of the amino ketone molecules can chelate the iron atoms at the oxide film surface to form rings (Figure 2). The high stability of the chelate ring structure enhances the adsorption of the inhibitor molecules on the steel surface;The aromatic groups that are connected to the carbonyl groups can serve as a barrier layer (steric hindrance effect) to separate the steel surface from the corrosive medium. The electronegativity of the aromatic groups can repel chloride ions away from the steel surface and therefore decrease the corrosion of chloride ions to the reinforced steel;Polyhydroxy groups were added to the nitrogen atoms of the amino ketone molecule to reduce the repulsive force between molecules, which is favorable to the densification and stability of the adsorption film.

## 3. Materials and Methods

### 3.1. Materials and Sample Preparation

A commercial corrosion inhibitor containing amino ketone molecules was used in this work. Samples of Q235 plain steel bars were cut into 50.0 × 25.0 × 5.0 mm for weight loss tests, and samples with a size of 10.0 × 10.0 × 10.0 mm for electrochemical experiments were embedded into epoxy resin and exposed to the electrolyte. All the samples were polished with emery papers (grade 100 to 2000), then degreased with acetone washing, and finally cleaned ultrasonically in ethanol and water.

### 3.2. Weight Loss Tests

After drying in a desiccator and weighing, the samples were immersed in a 400 mL solution containing 3.5% of NaCl and different amounts of the inhibitor (0, 1, 2, 3, and 4%) for 7 days (Figure 3). Afterwards, they were washed first with water. Then they were washed via a pickling solution prepared at a concentration ratio of 1:1:10 of water, hydrochloric acid, and C_6_H_12_N_4_(hexamethylenetetramine). Finally, the samples were ultrasonically cleaned with alcohol, dried with a desiccator, and weighed using an analytical balance. An average weight loss for each test group was calculated with values from three parallel samples. 

### 3.3. Open Circuit Potential (OCP) Test

The prepared working electrode was soaked in a saturated calcium hydroxide solution containing the rust inhibitor and sodium chloride for about 30 min, then the open-circuit potential test was carried out, and the open-circuit potential was stable when the potential change value was less than 2 mV within 5 min (300 s).

The specific test steps are as follows: open powersuit—new—opencircuit—Ecorr—Name of—reference electrode—SCE(KCl)—Finish.

### 3.4. Electrochemical Tests

The experiments were conducted using a PARSTAT 2273 Potentiostat/Galvanostat (AMETEK, Inc., Berwyn, PA, USA) in different solutions at 298 ± 2 K. The equipment consisted of a working electrode with the steel sample, a counter electrode with platinum foil, and a SCE reference electrode was connected with a three-electrode system (Cf. Figure 4).

The working electrode was immersed in different simulated solutions for 0.5 h before the electrochemical tests. The open-circuit potential (OCP) was recorded afterward at a steady state (potential value change below 2 mV in 300 s). The AC frequency for the tests was in the range of 10^5^ to 10^−2^ Hz with a peak-to-peak sine wave of 10 mV as the excitation signal. For the potentio dynamic polarization tests, the potential was scanned with a rate of 0.5 mV/s for the range of −250 to +250 mV (versus OCP). All the collected electrochemical data were analyzed with the software PowerSuite V2.5.0 and ZSimpWin Win10 version 3.60 [28]. There were four groups of tests in this work.

(1) The samples were immersed in 400 mL saturated calcium hydroxide (Ca(OH)_2_) solution containing 3.5% NaCl and different amounts of the inhibitor (0, 1, 2, 3, and 4%); 

(2) The concentration of NaCl added to the saturated Ca(OH)_2_ solution with/without 2% inhibitor was 2, 3.5, or 5%;

(3) The pH value of the saturated Ca(OH)_2_ solution containing 3.5% NaCl and 2% inhibitor was adjusted with sodium bicarbonate (NaHCO_3_) to the range of 9.3–12.3;

(4) The temperature of the solution listed above was controlled with a thermostat water bath at 20, 25, 30, 35, or 40 °C.

### 3.5. Surface Microscopic Analysis

The samples were immersed in SCPS containing 3.5% NaCl with and without 4% inhibitor for 24 h. Subsequently, each sample was dried at 20 °C after removal from the SCPS. The effect of corrosion inhibitor on the surface morphology of carbon steel was observed by optical microscope.

## 4. Results and Discussion

### 4.1. Weight Loss Tests

The weight loss data were used to calculate the average corrosion rates (v, g/m^2^⋅h) with Equation (1), and the inhibition efficiencies (*IE_w_*) against the exposure time were determined with Equation (2) [28]:
(1)$$V=\frac{W_{0}-W_{1}}{s\cdot t}$$
(2)$$IE_{w}=\frac{V_{0}-V_{1}}{V_{0}}\times 100\%$$
where, v_0_ and v_1_ are the corrosion rates with and without the inhibitor, respectively.

The results of the weight loss tests are listed in Table 1. It can be seen that with the increase in the inhibitor concentration, the steel corrosion rate decreased and the inhibition efficiency of the inhibitor was improved. A maximum *IE_w_* of 88.73% was achieved at the inhibitor concentration of 4%. This phenomenon could be explained by the good solubility and high adsorption capability (significantly higher speed for adsorption than desorption) of the organic inhibitor molecules. At a higher concentration, the molecules can form a more compact and complete protective film on the steel surface, which can prevent the local corrosion reactions more effectively.

### 4.2. OCP Test

The carbon steel electrodes were soaked in a saturated calcium hydroxide simulation pore solution containing 3.5% sodium chloride, one group containing the rust inhibitor, and another group without rust inhibitor. The open-circuit potential was measured after a certain period of time, and the results are shown in Figure 5. They showed that the carbon steel electrode forms a passivation film slowly in the solution without rust inhibitor, and it took more than 1 h for the open-circuit potential to be stable. After adding sodium chloride, the potential immediately shifts and moves to a negative value. The carbon steel electrode in the solution with rust inhibitor reached a stable potential in about 30 min, and the potential is higher than that of the blank group without rust inhibitor. It indicated that the addition of rust inhibitor promotes the appearance of passivation film on the surface of carbon steel.

### 4.3. Effect of the Inhibitor Concentration

The effects of the inhibitor concentration on the impedance behavior of steels in 3.5% NaCl solution are shown in Figure 6. The information on |Z| at 10^−2^ Hz are provided in Table 2. It can be seen from the Nyquist plots in Figure 6 that the impedance spectra diameter in the inhibitor solution was significantly higher than in the absence of the inhibitor. In addition, it increased with higher inhibitor concentrations. The difference in the impedance arc at inhibitor concentrations of 1 and 2% was relatively small. However, an obvious increase in the electrochemical resistance was observed for specimens with the inhibitor of 3%. This indicated that the corrosion reaction of the carbon steel surface was effectively decreased [29]. The results are consistent with Zhao’s [30] and Xu’s [31] findings.

The minimum error and the most accurate system were determined by comparing the fitted values of different equivalent circuits [1,5,28], as shown in Figure 7. R_s_ represents the solution resistance between the working and reference electrodes. R_f_ is the resistance for the film formed at the surface of copper. R_ct_ is defined as charge transfer resistance in the steel corrosion process. Capacitor C_1_ represents the capacitance of the membrane (C_f_) in the corrosion process, which originates mainly from the dielectric function of the surface film (the inhibitor film and/or the corrosion products). C_2_ represents the capacitance of the double-layer (C_dl_) [9].

The fitted results are shown in Table 3. The inhibition efficiency in the EIS tests can be calculated via Equation (3):
(3)$$IE\%=\frac{R_{ct}-R_{ct}^{0}}{R_{ct}}\times 100\%$$
where R_ct_ and R_ct_^0^ represent the charge transfer resistances with and without inhibitors, respectively. In comparison with the inhibitor-free cases, the corrosion resistance of steel samples under inhibitor-containing conditions was significantly higher. Moreover, an obvious increase in the inhibition efficiency with the inhibitor concentration was detected. From Table 3, it can be seen that *R_s_
* and *R_ct_* significantly increased with the addition of corrosion inhibitors. On the contrary, the values of *C*_1_ and *C*_2_ show a decreasing trend. The IE was increased by nearly 80% when adding 1% corrosion inhibitors. As the concentration increased to 4%, the IE reached 89.07%. It can be concluded that the inhibitor improves inhibition performance. This can be attributed to the corrosion resistance ability of the protective inhibitor film formed with molecules adsorbed on the steel surface, which can cover the activation area of the steel surface and effectively protect the steel from chloride ion-induced corrosion. The protective film became more compact and complete with the increase in inhibitor concentration, so the corrosion resistance was enhanced. This is consistent with the results of the weight loss measurements. In addition, the fitted data in Table 3 show a pattern almost similar to that of the experimental results from the R(C(R(CR))) equivalent circuit (Cf. Figure 6).

### 4.4. Effect of the NaCl Concentration

The Nyquist plots and Bode plots of the steel electrodes immersed in the simulated concrete pore solutions with different NaCl concentrations are shown in Figure 8, and the polarization curves are shown in Figure 9. In addition, |Z| at 10^−2^ Hz is shown in Table 4. Figure 8 shows that with the increase in NaCl concentration, the steel corrosion became more significant. After the addition of 2% inhibitor, the impedance spectra diameter of the steel electrode turned to a higher value than that of the blank groups with different concentrations of NaCl. This suggests an excellent corrosion resistance of the inhibitor for steels at different NaCl concentrations. Based on the fitted data, the inhibition efficiency according to the Equation (3) was calculated to be 65.62, 80.06, and 66.30% at NaCl concentrations of 2.0, 3.5, and 5.0%, respectively.

The cathodic polarization curves in Figure 9 indicate that the inhibitor does not affect the cathodic reaction mechanism. Furthermore, the breakdown potential of the inhibitor-containing solution is obviously higher than that of the inhibitor-free solution, and the passivation current density is lower in the presence of the organic inhibitor. This is because the protective film of inhibitor molecules on the carbon steel surface can repair the defects and pores of the steel passive film, and therefore inhibit the occurrence of steel corrosion. The results in Table 5 were obtained by fitting the linear polarization of carbon steel electrodes in simulated corrosion fluids with different concentrations of sodium chloride with 2% inhibitor and without inhibitor (Figure 9). Ecorr and icorr, based on Figure 9, are shown in Table 5, which indicates that the icorr of carbon steel in the solution containing rust inhibitors is lower than that of the blank group without rust inhibitor. It also indicates that rust inhibitors adsorb on the surface of steel bars and effectively prevent chloride ion corrosion, thereby improving the corrosion resistance of steel bars. The inhibition efficiency can be calculated via Equation (4):
(4)$$\eta =(i_{corr}-i_{corr}\prime )/i_{corr}$$
where icorr and icorr’ represent the corrosion current density of the blank group without inhibitor and the 2% inhibitor group, respectively. Calculation results show that the inhibition efficiency is 45.5, 91.2, and 64.1% at NaCl concentrations of 2.0, 3.5, and 5.0%, respectively. The result of the tafel polarization curve has a similar changing tendency as the result of EIS.

### 4.5. Effect of the pH Value

The strong alkali environment is beneficial to maintaining the stability of a passivation film. Related studies have shown that a steel bar can be completely passivated in a strong alkaline environment when the pH value is greater than 11.5. When the pH value near the steel bar reaches 9.88–11.5, even if there is no chloride ion near the steel bar, the film will also dissolve due to instability. It can be seen from Figure 10 that the impedance spectra diameter of the steel electrode increased with the pH value, meaning that the alkaline environment is favorable for enhanced corrosion resistance of the steel. It was also found that both the steel dissolution and the oxygen reduction processes were restricted in the presence of the inhibitor at all pH conditions. In addition, as shown in Figure 11, the inhibition efficiency first increased and decreased again afterwards with the increase in the pH value. The highest corrosion resistance value was observed at a pH value of 11.3, which could be attributed to the formation of complete passive film on the steel bar surface under this pH value condition.

### 4.6. Effect of the Temperature

The environments of the reinforced concrete structure worked in are different, such as in the different temperatures, so the simulated concrete pore solution with different temperatures were designed in this experiment. In order to study the performance of the rust inhibitor at different temperature, the electrochemical impedance spectra of carbon steel electrodes in saturated calcium hydroxide simulated pore solutions containing 2% rust inhibitor and 3.5% mass fraction sodium chloride at temperatures of 20, 25, 30, 35, and 40 °C, with one blank group without inhibitor set up for comparison.

The results (Cf. Figure 12) revealed a good corrosion resistance of the inhibitor for carbon steel electrode in 3.5% NaCl solution in the temperature range from 20 to 40 °C. The efficiency under different temperatures followed an order of 25 > 35 > 40 > 20 > 30 °C and the maximum inhibition efficiency of 81.32% was achieved at 25 °C. When the temperature is below 25 °C, as the temperature rose, the corrosion inhibitor molecules were more active than the corrosive medium and moved to the electrode surface first, maintaining the stability of the passivation film. When the temperature exceeded a certain range, the thermal movement of the corrosive medium was intensified, and its activity exceeded the activity of the corrosion inhibitors, first reaching the electrode surface and quickly passing through the phase interface film, then penetrating and diffusing into the metal substrate, resulting in increased membrane capacitance, which prevented the corrosion products from forming a dense film.

### 4.7. SEM Analysis

The surface morphology of carbon steel immersed in the blank solution and corrosion inhibitor solution after seven days have been enlarged 5000 times, as shown in Figure 13. The observed white snowflake substance is caused by the adsorption and precipitation of ${Ca(OH)}_{2}$ on the surface of the carbon steel. It is evident that the surface morphology of samples before and after adding the corrosion inhibitor is different. The surface of the carbon steel in the new solution is rough and uneven, and a large number of pitting pits and rust marks appear. This is an indication that the existence of chloride ions accelerates the corrosion of the reinforcing bar surface. However, the surface of the carbon steel added to the inhibitor solution is smooth, and there is no sign of corrosion. The surface is slightly white, and a layer of film appears. It shows that the addition of the corrosion inhibitor can create an adsorption film on the surface of the reinforcing bar, to prevent the corrosion by chloride ions on the reinforcement. Hence, the corrosion inhibitor can give excellent protection and repair effects on the reinforcement bar.

## 5. Conclusions

The corrosion inhibition performance of the new type of organic inhibitor containing amino ketone molecules under different conditions was studied and the conclusions aresummarized as follows:With the increasing content of the corrosion inhibitor in the concrete pore solution, the pitting potential (E_pit_) and the charge transfer resistance (R_ct_) increased, while current density (i_corr_) and the double layer capacitance (Q_dl_) showed the opposite trend;The corrosion inhibitors can be adsorbed on the surface of carbon steel to form a film, which can effectively reduce the pitting corrosion of the steel bar caused by harmful substances such as chloride;Alkaline media was found to be favorable for improving the corrosion resistance of steels. When the pH of the solution is 11.3 and the temperature is 25 °C, the rust inhibition effect is best.

## Figures and Tables

**Figure 1 materials-17-02168-f001:**
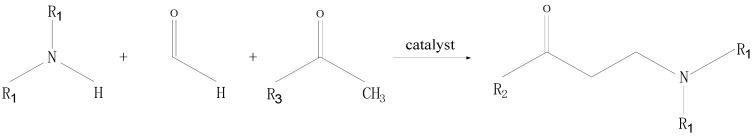
Synthesis process of the amino ketone inhibitor.

**Figure 2 materials-17-02168-f002:**
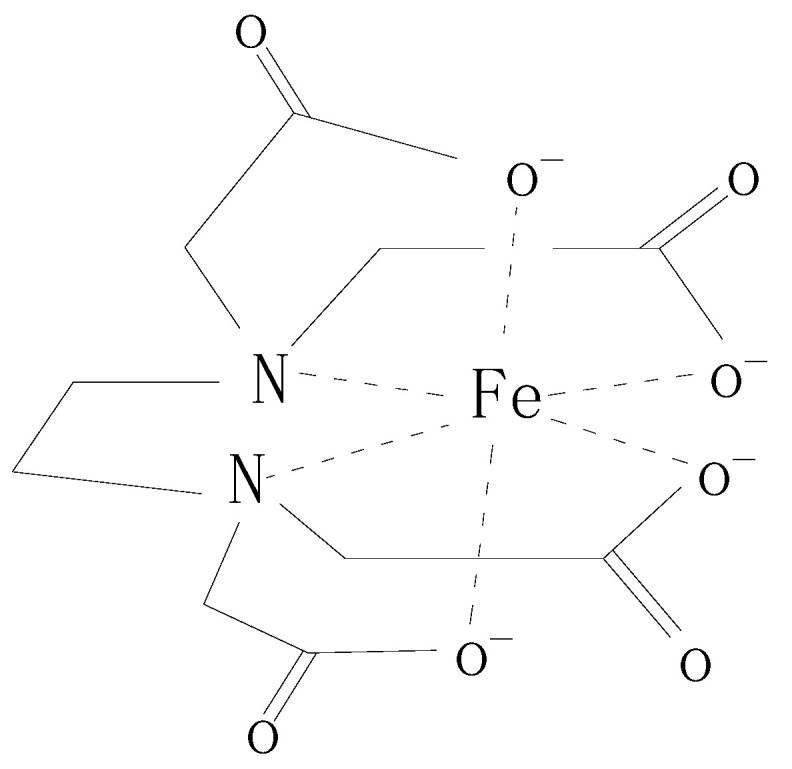
The chelate ring structure of inhibitor molecule with steel surface ion atom.

**Figure 3 materials-17-02168-f003:**
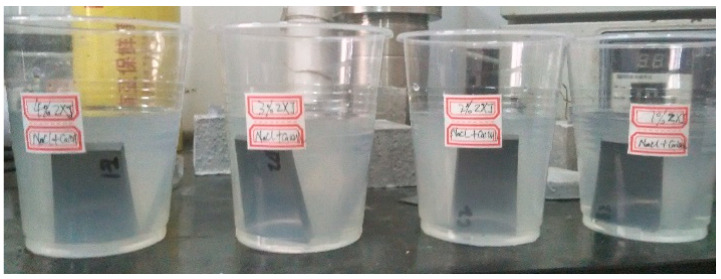
Picture of weight loss tests.

**Figure 4 materials-17-02168-f004:**
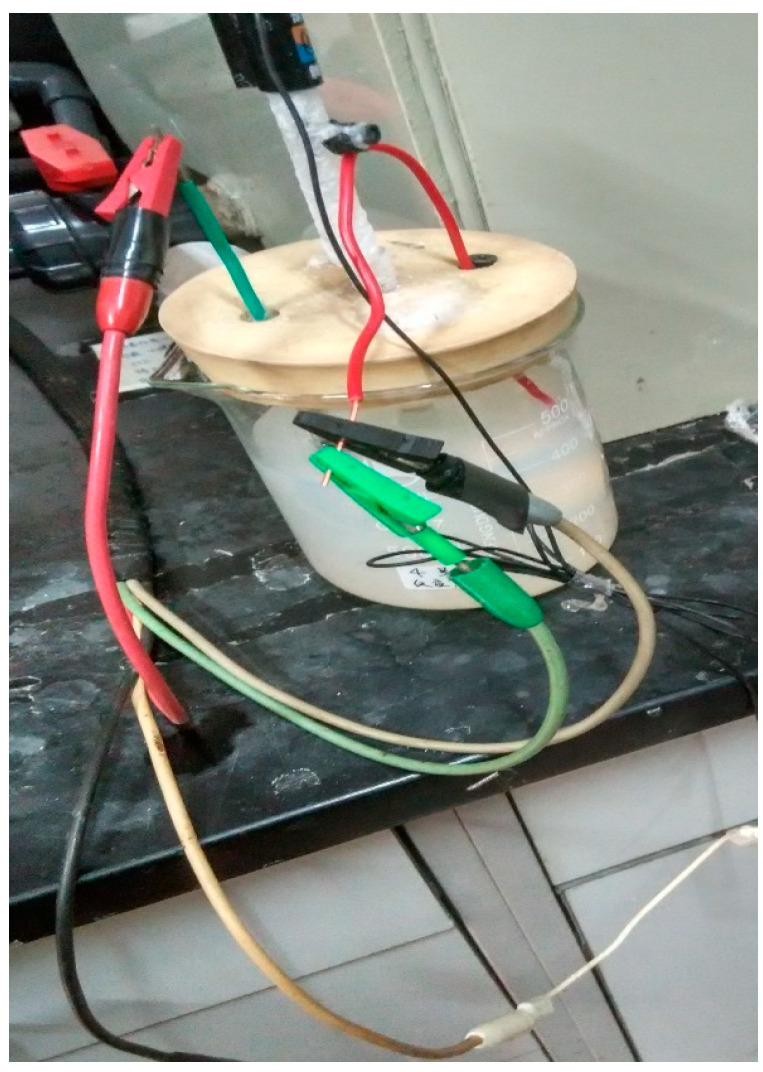
Picture of the three-electrode cell.

**Figure 5 materials-17-02168-f005:**
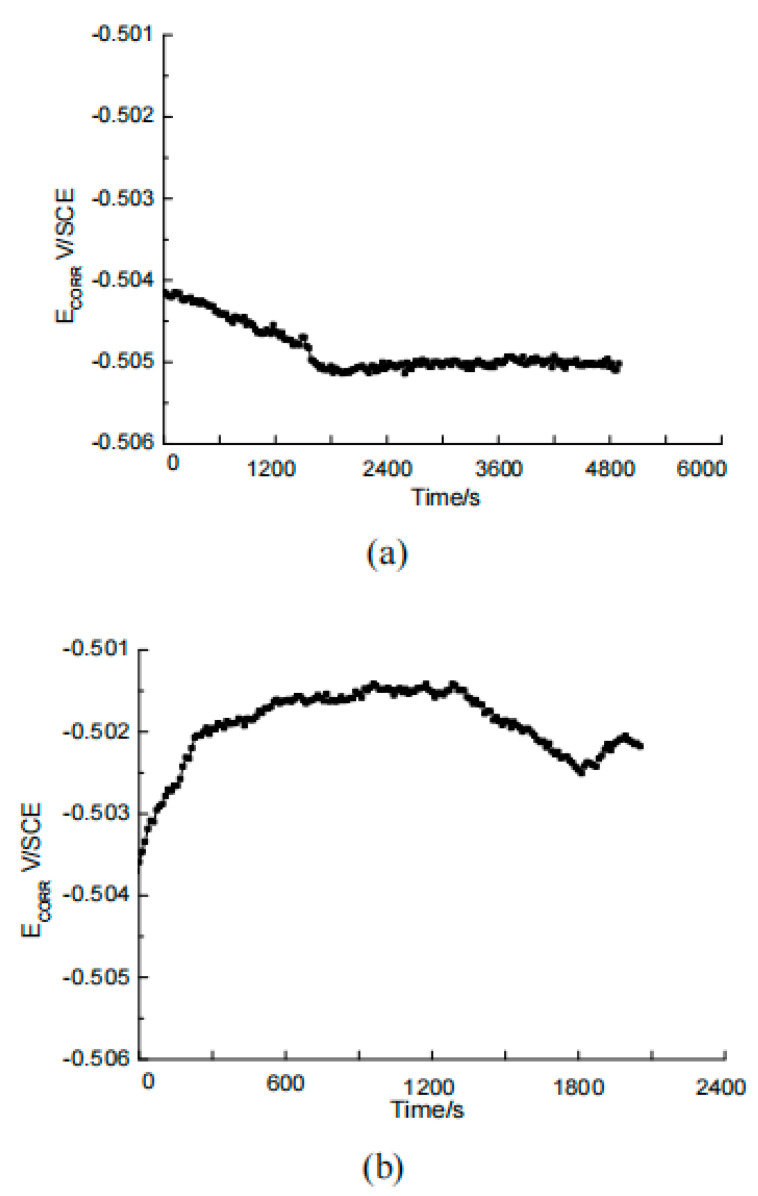
OCP test of blank group (**a**) and rust inhibitor group (**b**).

**Figure 6 materials-17-02168-f006:**
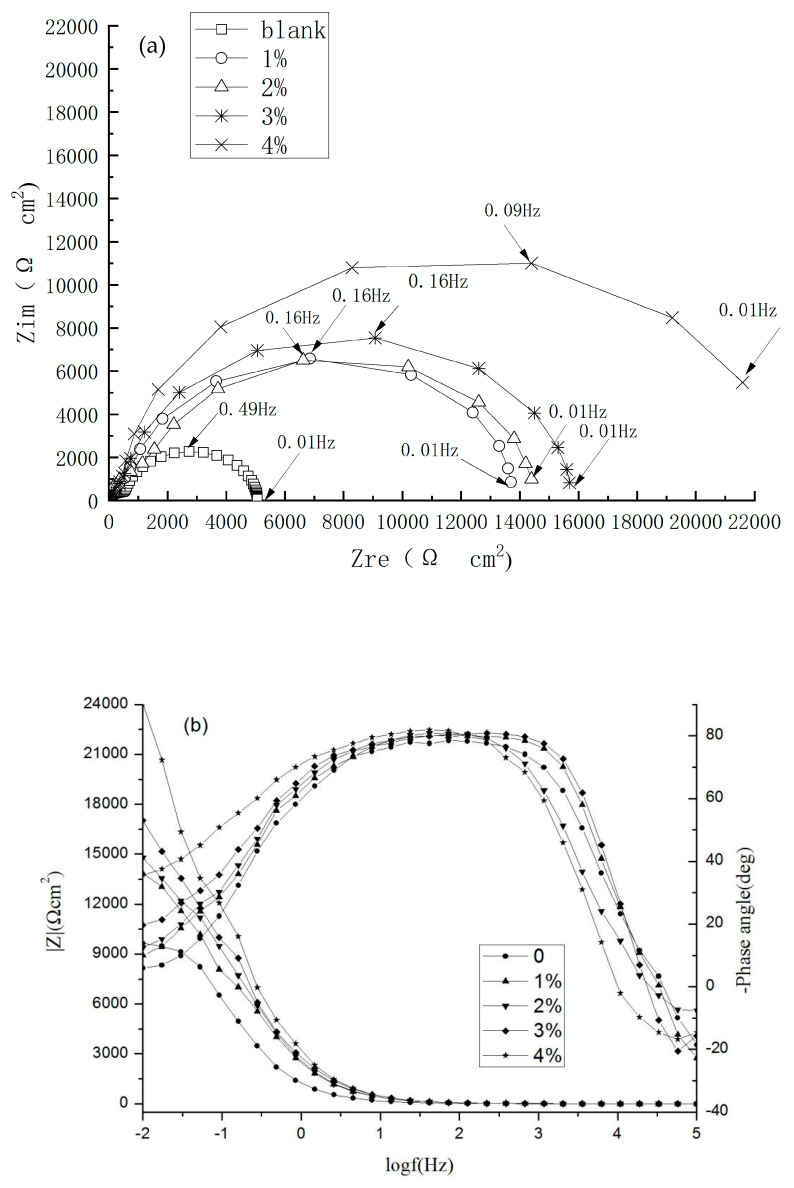
Nyquist plots (**a**) and Bode plots (**b**) of steels in solutions with different inhibitor concentrations.

**Figure 7 materials-17-02168-f007:**
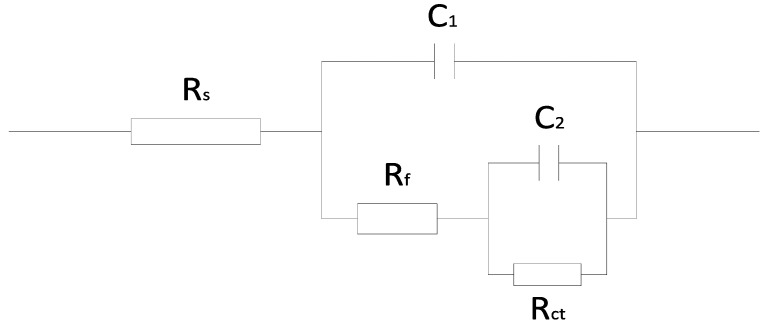
The equivalent circuit model applied to fit the EIS experimental data.

**Figure 8 materials-17-02168-f008:**
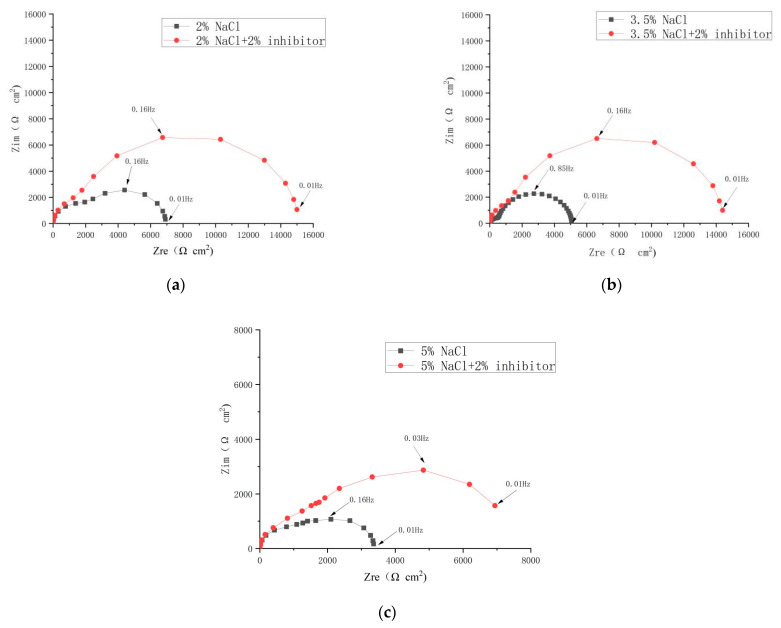
Nyquist plots (**a**–**c**) and Bode plots (**a**’–**c**’) of the steel in solutions with different NaCl concentrations.

**Figure 9 materials-17-02168-f009:**
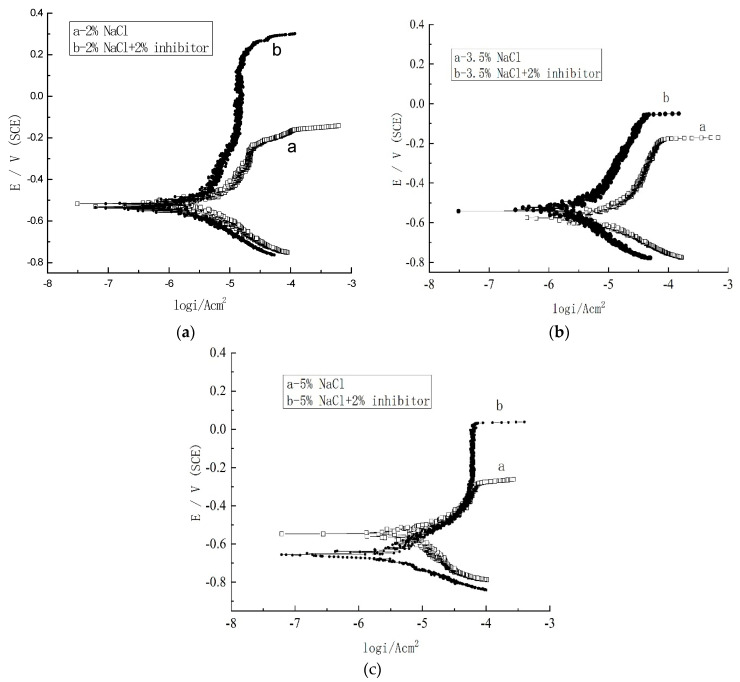
(**a**–**c**) The polarization curves of the steel in solutions with different NaCl concentrations.

**Figure 10 materials-17-02168-f010:**
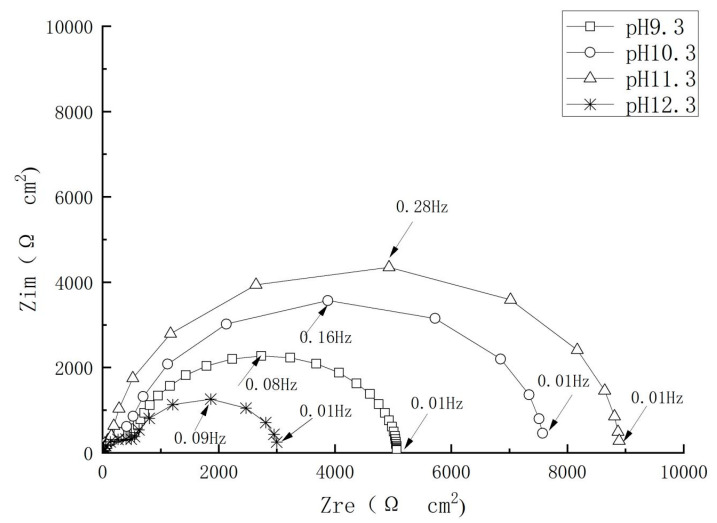
Nyquist plots of the steel in inhibitor-containing solution with different pH values.

**Figure 11 materials-17-02168-f011:**
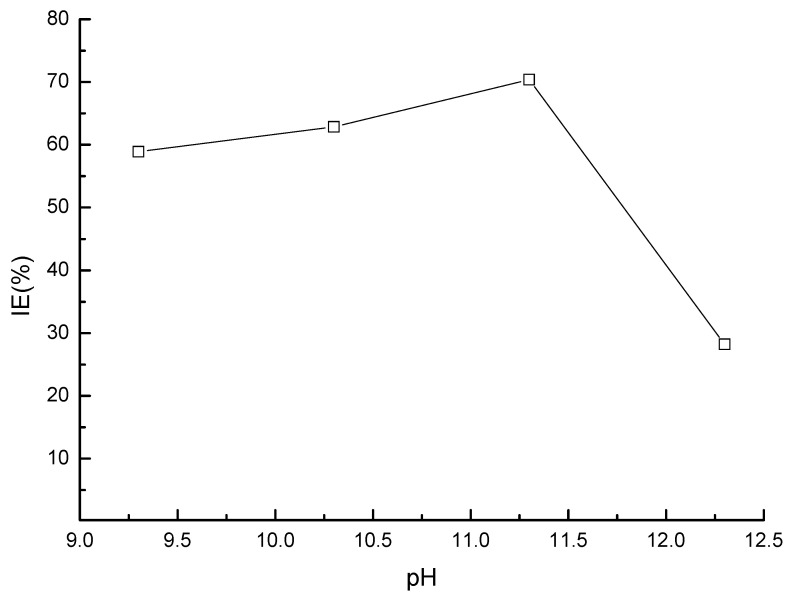
The inhibition efficiency of steel in solution of different pH values.

**Figure 12 materials-17-02168-f012:**
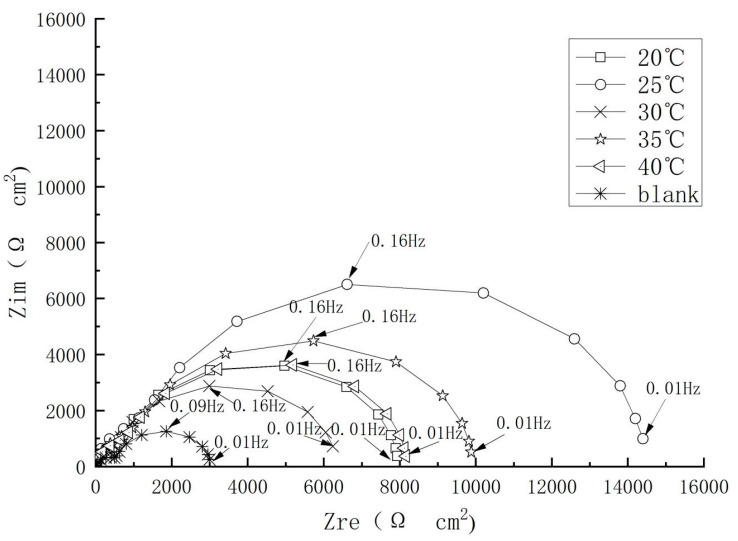
Nyquist plots of the steel in solution of different temperatures.

**Figure 13 materials-17-02168-f013:**
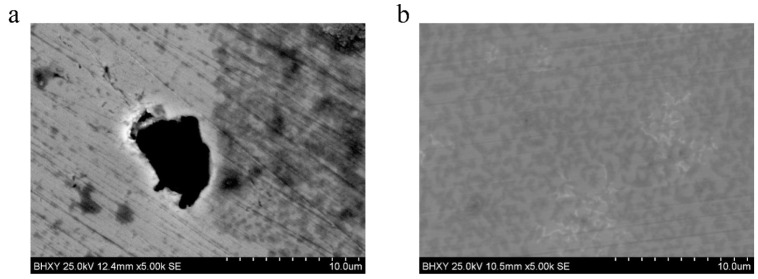
SEM images comparison of surface morphology of carbon steel experiment block after 7 days immersion, (**a**) blank group, (**b**) experiment group with inhibitor.

**Table 1 materials-17-02168-t001:** The average corrosion rate and the inhibition efficiency of the inhibitor calculated with data from weight loss tests.

Inhibitor Concentration	v (gm^−2^ h^−1^)	IE_w_ (%)
0	0.0403	/
1%	0.0089	77.73
2%	0.0077	80.85
3%	0.0060	84.91
4%	0.0045	88.73

**Table 2 materials-17-02168-t002:** |Z| at 10^−2^ Hz.

**Inhibitor Concentration**	0	1%	2%	3%	4%
**|Z|/(Ωcm^2^)**	9641.3	13,846	14,791	17,023	24,073

**Table 3 materials-17-02168-t003:** Fitted results of equivalent circuit elements in concrete simulation.

Conc. (%)	R_s_ (Ωcm^2^)	C_1_ (μFcm^2^)	R_f_ (Ωcm^2^)	C_2_ (μFcm^2^)	R_ct_ (Ωcm^2^)	IE (%)
0	2.867	60.8	643.6	614.7	2381	/
1%	2.464	29.7	2909	72.4	11550	79.39
2%	1.097	31.8	1860	54.0	11940	80.06
3%	1.346	26.3	1835	35.0	13870	82.83
4%	2.018	24.2	1249	37.5	21780	89.07

**Table 4 materials-17-02168-t004:** |Z| at 10^−2^ Hz.

	**NaCl**	2%	3.5%	5%
**Inhibitor**				
**0%**		6916.911	3015.345	3368.133
**2%**		15,008.92	8890.527	7118.817

**Table 5 materials-17-02168-t005:** Ecorr and Icorr of the steel at different NaCl concentrations.

Solutions with Different NaCl and Inhibitor Concentrations	Ecorr (V vs. SCE)	Log (Icorr) (A·m^2^)	Icorr (μA/cm^2^)	H (%)
2%NaCl	−0.792	−5.586	2.594	
2%NaCl + 2%Inhibitor	−0.526	−5.850	1.413	45.5
3.5%NaCl	−0.548	−5.358	4.385	
3.5%NaCl + 2%Inhibitor	−0.524	−6.416	0.384	91.2
5%NaCl	−0.566	−5.478	3.327	
5%NaCl + 2%Inhibitor	−0.635	−5.923	1.194	64.1

## Data Availability

Data are contained within the article.

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
