# Peer review of "Rust Prevention Property of a New Organic Inhibitor under Different Conditions"

_materials, 2024, doi:10.3390/ma17092168_

Round 1
Reviewer 1 Report
Comments and Suggestions for Authors
In this study Rust prevention property of a new protective organic inhibitor under different conditions, the authors present the corrosion resistance properties of a new type of environment-friendly organic inhibitor containing amino ketone molecules. The efficiency of the inhibitor was evaluated using electrochemical (EIS, PD) and weight loss methods.
Some major revisions must be made before publication.
1. In my opinion the title must be changed because in improper to say protective organic inhibitor. The role of an inhibitor is to be protective (from the definition of inhibitor).
2. The phrase “Cathodic protection is an effective way to suppress steel corrosion” needs to be rewritten. To suppress means zero corrosion rate that is impossible. Maybe to say decrease or to reduce.
3. Reference [28] are wrong. The authors indicated Ma et al as authors and at reference is [28] Fubin M., Weihua L., Huiwen T., et al. The Use of a New Thiadiazole Derivative as a Highly Effi-cient and Durable Copper Inhibitor in 3.5% NaCl Solution [J]. International Journal of Electrochemical Sci-ence, 2015, 10(7): 5862-5879. Please carefully check the references.
4. The synthesis of the inhibitor is very little described and must be completed with the details of the synthesis reaction (reactant concentrations, temperature, catalyst type, molecular structure and so on). – Figure 1 is too generally.
5. In the paper title the authors mentions “new protective organic inhibitor“ and at materials and methods the authors say: “A commercial corrosion inhibitor containing amino ketone molecules was used in this work”. So that the inhibitor is new, synthesized or is a commercial product that is not new? Please correct and give more details about the used inhibitor.
6. C6H12N4 is better to write the IUPAC name.
7. Figures 3 and 5 is not necessary in the paper, because are well known.
8. Maybe is better that the authors to put the OCP diagrams in the paper (from my point of view the 0.5 hours before the electrochemical tests is a short period that the system evaluate to reach the corrosion potential).
9. The authors to not mentions the analysis of the inhibitor used and some characteristics of this (FTIR maybe).
10. The authors say that “A maximum IEw of 80.91% was achieved at the inhibitor concentration of 4%”. From Table 1 the value of inhibitor efficiency was 88.73%
11. The authors say: “At a higher concentration, the molecules can form a more compact/complete protective film on the steel surface, which can prevent the local corrosion reactions more effectively”. How did the authors reach this conclusion? the paper does not present the morphology of the steel surfaces before and after corrosion (missing optical microscopy, scanning electron microscopy or atomic force microscopy images). Please complete this study with images about the morphology of the steel surfaces before and after corrosion tests
12. The evaluation of inhibitor efficiency was evaluated only for first group of tests from four groups that authors presented? The conclusions for one group of tests are not relevant. Please give values about corrosion efficiency of the inhibitors for others parameters (temperature, pH).
13. Figure 7 are two figures? what does each represent? The authors don’t mention this.
14. The authors mention “the diameter of the steel electrode in the inhibitor solution was significantly” that is wrong. The diameter of the steel electrode is the same for all sample and is not a diameter is working electrode surface. Authors probably want to say about the diameter of the semicircle obtained from impedance spectra. Please rewrite the sentence more carefully.
15. In addition, it increased with higher inhibitor concentrations. Is wrong because from the impedance spectra the trend is 2%, 1%, 3% and 4%.
16. “The difference in the impedance arc at inhibitor concentrations of 1 and 2% was relatively small”. Relatively small is not proper because the difference is around 60-70 ohm ·cm2 that is not a small difference and contradicts the statement from point 14.
17. “The minimum error and the most accurate system were determined by comparing the fitted values of different equivalent circuits, as shown in Fig. 8.” Figure 8 present only one equivalent circuit used for fitting EIS data. What are the other circuits that the authors refer to? Authors should complete or be more specific.
18. Figure 9 (a, b c) must compress to one figure. Are only 2 diagrams on graphic and is not proper to compare the influence of the NaCl concentrations. More than that all data are in range of 0 – 16.000 Ω·cm2. Please put all data in only one figure.
19. The parallel cathodic polarization curves in Fig. 10. Figure 10 is not a parallel cathodic polarization. Please carefully check the polarization curve.
20. The authors must complete the study with values obtained from potentiodynamic polarization curves, such as: corrosion potential, corrosion current density, Tafel parameters, polarization resistance and corrosion rate to better understand the efficiency of the inhibitor use for testing and to sustain the affirmations from the study. More than that, the authors must compare the results (polarization resistance and corrosion rate) obtained from both testing methods: electrochemical impedance spectroscopy and potentiodynamic polarization curves in order to have o conclusion about the inhibitor efficiency.
21. The authors say: “The highest corrosion resistance value was observed at a pH value of 11.3 “, but from results presented in Figure 12 the highest efficiency is for pH = 12.3 (highest polarization resistance that means lowest corrosion rate and best inhibitor efficiency). Please check the values and make a viable conclusion from the presented results.
22. Results obtained from figures 12 and 13 are in contradictory. Please check the data.
23. “The impedance spectra of the inhibitor-free solution at different temperatures are very close to each other, so only one of them is shown here for comparison.” The sentence is wrong because from the values obtained from figure 14 the values obtained only at 20°C and 40°C are approximatively the same. The difference between results obtained at 30°C, 20°C and 25°C is about 2000 ohms·cm2, so is not “very close to each other”. Please check the results and their correct interpretation carefully.
24. The authors say” good corrosion resistance of the inhibitor for copper”. The inhibitor was evaluated for steel not for copper (at Materials and methods the authors mentioned: “Samples of Q235 plain”. More than that, the authors do not mention the composition of steel used for tests. Please check carefully, again, the affirmation from the paper.
25. The authors say: “It was found that the molecular motion was stronger with the increase of the temperature in a specific range”. what determinations and experimental results or bibliographical references were the basis of the formulation of the phrase? Please justified your expression.
26. The results obtained and presented in the 4. Results and discussion part must be shared with other experimental results obtained in works with the same research theme. please complete the bibliography with such references and make comparisons in the text of the work with various values of the corrosion speed obtained for the inhibitors.
27. The conclusions must be rewrite after the authors complete the paper with tests mentioned before. The present conclusions are not supported by the data presented in this study.

Author Response
Dear Professor,
We are very grateful to you that give us an opportunity to revise and resubmit our manuscript. We also thank the reviewers for carefully reviewing the manuscript and providing extremely helpful feedback. We have made great efforts to revise the manuscript.
Reviewers’ comments are numbered and appear verbatim in blue words, authors responses appear immediately following each comment in black words.
Notice that all the numbers of lines mentioned below aim at the new submitting unless special statement. After completing revision, we submit two versions of the manuscript, i.e., one is the original version with revision mark (manuscript marked), where revision parts are marked in yellow color; another is the ultimate version (manuscript ultimate).
Reviewer #1: In this study Rust prevention property of a new protective organic inhibitor under different conditions, the authors present the corrosion resistance properties of a new type of environmentfriendly organic inhibitor containing amino ketone molecules. The efficiency of the inhibitor was evaluated using electrochemical (EIS, PD) and weight loss methods.
Some major revisions must be made before publication.
- In my opinion the title must be changed because in improper to say protective organic inhibitor. The role of an inhibitor is to be protective (from the definition of inhibitor).
Thank you very much for your suggestion, and "protective" in the title has been deleted.
- The phrase “Cathodic protection is an effective way to suppress steel corrosion” needs to be rewritten. To suppress means zero corrosion rate that is impossible. Maybe to say decrease or to reduce.
Thanks for your suggestion. We have change “suppress” to “decrease” in the revised paper.
- Reference [28] are wrong. The authors indicated Ma et al as authors and at reference is [28] Fubin M., Weihua L., Huiwen T., et al. The Use of a New Thiadiazole Derivative as a Highly Effi-cient and Durable Copper Inhibitor in 3.5% NaCl Solution [J]. International Journal of Electrochemical Sci-ence, 2015, 10(7): 5862-5879. Please carefully check the references.
I am so sorry for the formatting error. Reference [28] has been corrected in the revised paper as Ma F., Li W., Tian H., et al. The Use of a New Thiadiazole Derivative as a Highly Efficient and Durable Copper Inhibitor in 3.5% NaCl Solution[J].International Journal of Electrochemicalence, 2015, 10(7):5862-5879.
4.The synthesis of the inhibitor is very little described and must be completed with the details of the synthesis reaction (reactant concentrations, temperature, catalyst type, molecular structure and so on). – Figure 1 is too generally.
Thanks for your suggestion. We added the synthesis of the inhibitor to 2. Synthesis of the inhibitor. The organic rust inhibitor used in the test contains the amino ketone molecules made by the chemical reaction of ethanol, dimethylamine, formaldehyde, acetophenone and other substances, and its molecular generation formula is shown in Fig. 1.
5.In the paper title the authors mentions “new protective organic inhibitor “and at materials and methods the authors say: “A commercial corrosion inhibitor containing amino ketone molecules was used in this work”. So that the inhibitor is new, synthesized or is a commercial product that is not new? Please correct and give more details about the used inhibitor.
Thanks for your suggestion. The rust inhibitor studied in this paper is a newly developed new material, which has not been put into the market. In this paper, its performance and the best applicable conditions are studied experimentally.
- C6H12N4 is better to write the IUPAC name.
Thanks for your suggestion. It has been corrected C6H12N4 (hexamethylenetetramine) in the revised paper.
7.Figures 3 and 5 is not necessary in the paper, because are well known.
Thanks for your suggestion. Figures 3 and 5 have been erased in the revised paper.
8.Maybe is better that the authors to put the OCP diagrams in the paper (from my point of view the 0.5 hours before the electrochemical tests is a short period that the system evaluate to reach the corrosion potential).
Thanks for your suggestion. We have added the OPC test to 3.3 and the result was showed in 4.2 in the revised paper.
9.The authors to not mentions the analysis of the inhibitor used and some characteristics of this (FTIR maybe).
Thanks for your suggestion. The characteristics of inhibitor used were mentioned in 2. Synthesis of the inhibitor. Four characteristics were introduced.
10.The authors say that “A maximum IEw of 80.91% was achieved at the inhibitor concentration of 4%”. From Table 1 the value of inhibitor efficiency was 88.73%
I am so sorry for the mistake and it has been corrected “A maximum IEw of 88.73% was achieved at the inhibitor of 4%”in the revised paper.
- The authors say: “At a higher concentration, the molecules can form a more compact/complete protective film on the steel surface, which can prevent the local corrosion reactions more effectively”. How did the authors reach this conclusion? the paper does not present the morphology of the steel surfaces before and after corrosion (missing optical microscopy, scanning electron microscopy or atomic force microscopy images). Please complete this study with images about the morphology of the steel surfaces before and after corrosion tests
Thank you for your suggestion. SEM test has been added in the paper to verify the "forming a dense protective film on the surface of steel bars to prevent local corrosion of steel bars" mentioned in the paper.
- The evaluation of inhibitor efficiency was evaluated only for first group of tests from four groups that authors presented? The conclusions for one group of tests are not relevant. Please give values about corrosion efficiency of the inhibitors for others parameters (temperature, pH).
Thank you for your advice, and a detailed explanation of the results related to the effects of temperature and pH on rust inhibitors has been added in the paper.
13.Figure 7 are two figures? what does each represent? The authors don’t mention this.
Thanks for your suggestion. Two graphs from the same experimental data, but with different value ranges, have been merged into one graph.
- The authors mention “the diameter of the steel electrode in the inhibitor solution was
significantly” that is wrong. The diameter of the steel electrode is the same for all sample and is not a diameter is working electrode surface. Authors probably want to say about the diameter of the semicircle obtained from impedance spectra. Please rewrite the sentence more carefully.
Thank you very much and you are absolutely right. We have change “diameter of the steel electrode ” to “diameter of the semicircle obtained from impedance spectra”in the revised paper.
- In addition, it increased with higher inhibitor concentrations. Is wrong because from the impedance spectra the trend is 2%, 1%, 3% and 4%.
Thanks for your suggestion. From the figure of nyquist plots of steels in solutions with different inhibitor concentrations, we can find the impedance spectra with 1% and 2% inhibitor concentrations are close and there is no pattern. When Zre is less than 7000 probably, diameter of the semicircle obtained from impedance spectra with 2% is greater than 1%, but when Zre is more than 7000, the result is opposite.
16.“The difference in the impedance arc at inhibitor concentrations of 1 and 2% was relatively small”. Relatively small is not proper because the difference is around 60-70 ohm ·cm2 that is not a small difference and contradicts the statement from point 14.
I'm very sorry for my phrasing, the smaller difference in impedance arc said here at 1% and 2% inhibitor concentrations is relative to 3% and 4% inhibitor concentrations.
- “The minimum error and the most accurate system were determined by comparing the fitted values of different equivalent circuits, as shown in Fig. 8.” Figure 8 present only one equivalent circuit used for fitting EIS data. What are the other circuits that the authors refer to? Authors should complete or be more specific.
We appreciate very much for your careful and rigorous check. Figure 8 present one equivalent circuit often used for fitting EIS data .We haven‘t do tests of different equivalent circuit, because this equivalent circuit is cited in many references which states that this circuit is the most suitable. In addition, we annotated the references in the revised paper.
- Figure 9 (a, b c) must compress to one figure. Are only 2 diagrams on graphic and is not proper to compare the influence of the NaCl concentrations. More than that all data are in range of 0 – 16.000 Ω·cm2. Please put all data in only one figure.
Thanks for your suggestion. The rust inhibitor efficiency was obtained by comparing the impedance spectrum changes of the blank group without rust inhibitor and the group added rust inhibitor. Figure 9 (a, b c) respectively showed blank group and 2% inhibitor group with different NaCl concentrations. If figure 9 (a, b c) compress to one figure,it will be not clear enough. In addition, all data are in range of 0 – 16.000 Ω·cm2 and we can find electrochemical impedance spectra rough paths of blank group without inhibitor at different NaCl concentrations,and the same to 2% inhibitor group .The precise results are shown in Figure 11 Breakdown potential of the steel at different NaCl concentrations.
- The parallel cathodic polarization curves in Fig. 10. Figure 10 is not a parallel cathodic polarization. Please carefully check the polarization curve.
I am so sorry for the false statement and it has been corrected in the revised paper.
- The authors must complete the study with values obtained from potentiodynamic polarization curves, such as: corrosion potential, corrosion current density, Tafel parameters, polarization resistance and corrosion rate to better understand the efficiency of the inhibitor use for testing and to sustain the affirmations from the study. More than that, the authors must compare the results (polarization resistance and corrosion rate) obtained from both testing methods: electrochemical impedance spectroscopy and potentiodynamic polarization curves in order to have o conclusion about the inhibitor efficiency.
Thanks for your suggestions, detailed analysis of the relevant parameters in Table 2 has been added in the paper.
- The authors say: “The highest corrosion resistance value was observed at a pH value of 11.3 “, but from results presented in Figure 12 the highest efficiency is for pH = 12.3 (highest polarization resistance that means lowest corrosion rate and best inhibitor efficiency). Please check the values and make a viable conclusion from the presented results.
I am so sorry for the mistake. We checked the data seriously and the data graph was regenerated in the revised paper.
22.Results obtained from figures 12 and 13 are in contradictory. Please check the data.
Thanks for your suggestion. It was corrected in conjunction with the last one (suggestion21).
23.“The impedance spectra of the inhibitor-free solution at different temperatures are very close to each other, so only one of them is shown here for comparison.” The sentence is wrong because from the values obtained from figure 14 the values obtained only at 20°C and 40°C are approximatively the same. The difference between results obtained at 30°C, 20°C and 25°C is about 2000 ohms·cm2, so is not “very close to each other”. Please check the results and their correct interpretation carefully.
Thanks for your suggestion and I am sorry for the unclear description. What we want to express is that the test groups added rust inhibitor were set up separately at different temperatures, but the blank group without rust inhibitor only one group. Because we mainly study the rust prevention property of the inhibition affected by temperature. So, we modified the expressions in revised paper.
- The authors say” good corrosion resistance of the inhibitor for copper”. The inhibitor was evaluated for steel not for copper (at Materials and methods the authors mentioned: “Samples of Q235 plain”. More than that, the authors do not mention the composition of steel used for tests. Please check carefully, again, the affirmation from the paper.
Thanks for your suggestion. We changed “copper” to “carbon steel electrode” in the revised paper.
- The authors say: “It was found that the molecular motion was stronger with the increase of the temperature in a specific range”. what determinations and experimental results or bibliographical references were the basis of the formulation of the phrase? Please justified your expression.
Thanks for your suggestion. We revised the expression in the revised paper.
- The results obtained and presented in the 4. Results and discussion part must be shared with other experimental results obtained in works with the same research theme. please complete the bibliography with such references and make comparisons in the text of the work with various values of the corrosion speed obtained for the inhibitors.
Thank you for your suggestion. At present, there have been many researches on organic rust inhibitors, and the comparative study on the rust inhibition efficiency of the rust inhibitors studied in this paper compared with other rust inhibitors has been supplemented in the results discussion in the fourth part of this paper
- The conclusions must be rewrite after the authors complete the paper with tests mentioned before. The present conclusions are not supported by the data presented in this study.
Thanks for your suggestion. In conclusion, the variation law of rust inhibition efficiency of this new type of organic rust inhibitor under different conditions is summarized, and finally the most suitable conditions for its use are summarized(pH ,NaCl concentrations and temprature) ,which provides an important reference for its use in engineering.

Reviewer 2 Report
Comments and Suggestions for Authors
- Abstract, my recommendation is change “65.62%, 80.06%, and 66.30%” to “65.62, 80.06, and 66.30%” and the same recommendation for all manuscript
- Keywords: “EIS” try to use full name in the keywords
- 3. Materials and methods, include purification information or sentence “All materials were used as received”
- Figures s should move to supplementary materials or erase
- Manuscript has some types one example: 3.2. C6H12N4, correct according chemical rules
- Figures 4 and 5 should move to supplementary material
- Figure 7 has two pictures, but missing the information for bottom plot, top figures has information about blank, a, 2, 3 and 4%. Include information for bottom picture
- Manuscript has some interesting information but doesn’t have good discussion, improve this part
- Conclusion part needs to improve and include important results
Author Response
Dear Professor,
We are very grateful to you that give us an opportunity to revise and resubmit our manuscript. We also thank the reviewers for carefully reviewing the manuscript and providing extremely helpful feedback. We have made great efforts to revise the manuscript.
Reviewers’ comments are numbered and appear verbatim in blue words, authors responses appear immediately following each comment in black words.
Notice that all the numbers of lines mentioned below aim at the new submitting unless special statement. After completing revision, we submit two versions of the manuscript, i.e., one is the original version with revision mark (manuscript marked), where revision parts are marked in yellow color; another is the ultimate version (manuscript ultimate).
Reviewer #2: Comments and Suggestions for Authors
- 1.Abstract, my recommendation is change “65.62%, 80.06%, and 66.30%” to “65.62, 80.06, and 66.30%” and the same recommendation for all manuscript
Thanks for your suggestion. We have changed “65.62%, 80.06%, and 66.30%” to “65.62, 80.06, and 66.30%” and the same recommendation for all manuscript in the revised paper.
- 2.Keywords: “EIS” try to use full name in the keywords
Thanks for your kind suggestion. “EIS” in the keywords has been change full name “electrochemical impedance spectroscopy” in the revised paper.
- 3. Materials and methods, include purification information or sentence “All materials were used as received”
All the materials and methods mentioned in the paper are in conformity with the requirements of the test specification.
4.Figures s should move to supplementary materials or erase.
Thanks for your suggestion. I'd like to make sure if you're talking about Figure 3. It has been erased.
-5.Manuscript has some types one example: 3.2. C6H12N4, correct according chemical rules
I am so sorry. It has been corrected C6H12N4 (hexamethylenetetramine).
- 6.Figures 4 and 5 should move to supplementary material
Thanks for your suggestion. Figures 5 has been erased and the figures 4 has been modified. Figure 4 is picture of weight loss tests, including many materials such as samples of Q235 plain steel bars mentioned above and the solution containing 3.5% of NaCl and different amounts of the inhibitor used for weight loss tests.
- 7.Figure 7 has two pictures, but missing the information for bottom plot, top figures has information about blank, a, 2, 3 and 4%. Include information for bottom picture
Thanks for your suggestion. Two graphs from the same experimental data, but with different value ranges, have been merged into one graph.
-8. Manuscript has some interesting information but doesn’t have good discussion, improve this part
Thank you for your suggestions, and the analysis of relevant results has been improved in the paper.
- 9.Conclusion part needs to improve and include important results
Thanks for your suggestion. The important conclusions have been revised in the paper.

Round 2
Reviewer 1 Report
Comments and Suggestions for Authors
The paper can be published in present form
Author Response
Thank you very much for your support.